# Transcriptional Activity of Tumor Necrosis Factor Alpha Genes and Their Receptors in Patients with Varying Degrees of Coronary Artery Disease

**DOI:** 10.3390/ijms252313102

**Published:** 2024-12-05

**Authors:** Katarzyna Potyka, Józefa Dąbek

**Affiliations:** 1Faculty of Health Sciences in Katowice, Medical University of Silesia in Katowice, 40-635 Katowice, Poland; 2Department of Cardiology, Faculty of Health Sciences in Katowice, Medical University of Silesia in Katowice, 40-635 Katowice, Poland

**Keywords:** TNF-α, transcriptional activity, coronary artery disease

## Abstract

Coronary artery disease and its complications are one of the most common causes of morbidity and death worldwide. The aims of this study were to assess the transcriptional activity of the studied TNF-α genes and their receptors in patients with various degrees of coronary artery disease and in the control group, as well as to attempt to link it with the size of the left ventricular ejection fraction and the number of diseased coronary arteries. Taking into account the inclusion and exclusion criteria, a total of 240 people (100%) qualified for this study. For proper interpretation of the results of the molecular analyzes, the study group (240, 100%) was divided into a control group (C: *n* = 60; 25%), a group of patients with early coronary artery disease (W: *n* = 60; 25%), a group with stable coronary artery disease (S: *n* = 60; 25%), and a group of patients with acute coronary syndrome (ACS: *n* = 60; 25%). The transcriptional activity of the TNF-α genes and their receptors was assessed in peripheral blood mononuclear cells by a quantitative real-time polymerase chain reaction. The expression of the studied genes was inferred from the number of mRNA copies per 1 ug of total RNA. The analysis of the obtained results showed a significant increase in the transcriptional activity of the TNF-α gene with the severity of coronary artery disease, accompanied by a decrease in the activity of its receptor genes. Taking into account the number of affected coronary arteries and the size of the ejection fraction in the examined patients, there were no statistically significant differences in the transcriptional activity of the TNF-α receptor gene type I and II. The observed increase in the transcriptional activity of the TNF-α gene with a concomitant decrease in the activity of its receptor genes with the advancement of coronary artery disease, compared to the control group, may indicate their significant participation in the development and progression of the disease and constitute a useful marker in non-invasive, early diagnostics. In the patients of the study group, no changes in the transcriptional activity of the TNF-α genes and their receptors were demonstrated depending on the number of diseased coronary arteries and the size of the ejection fraction of the left ventricle.

## 1. Introduction

Coronary heart disease (CHD) is often equated with coronary artery disease (CAD). However, it is a broader concept, which in fact includes coronary artery disease (98%) but also other conditions (2%), causing myocardial ischemia [1].

Coronary artery disease and its complications are one of the most common causes of morbidity and death in the world. In Europe, it is the leading cause of death in people under 75 years of age [2]. Additionally, CAD is the cause of frequent hospitalizations, medical consultations, repeated diagnostics, and absences from work. The above-mentioned situations result in large financial outlays of the state for healthcare.

The cause of coronary artery atherosclerosis develops on the basis of inflammation, with a concomitant increase in prothrombotic activity and the release of proinflammatory cytokines, including interleukin 2 and interleukin 6 and tumor necrosis factor alpha. Atherosclerotic changes progress with age. However, the first atherosclerotic changes may occur already in the fetal period [3,4]. The inflammatory process is present in all stages of the development and progression of coronary atherosclerosis, and thus at every stage of the development of coronary artery disease. Among cytokines, the main mediator of an acute and chronic inflammatory response is tumor necrosis factor alpha. It has been reported that high levels of TNF-α are associated with an increased risk of cardiovascular disease. Physiologically, it has a highly pleiotropic effect, affecting practically every type of cell. It induces cellular responses starting from the induction of inflammatory gene expression programs through the stimulation of cell proliferation and differentiation to the activation of their programmed death: apoptosis and necroptosis [5]. In the cardiovascular system, TNF-α-activated signal transduction pathways play a crucial role in vascular dysfunction, CAD development and progression, and adverse cardiac remodeling following myocardial infarction and heart failure. Tumor necrosis factor alpha (TNF-α) has two receptors: tumor necrosis factor alpha type I receptor (TNFR1), also referred to as CD120a and p55, and tumor necrosis factor alpha type II receptor (TNFR2), also called CD120b and p75. TNF-α receptors are not specific. This means that they can also bind to other cytokines, such as lymphotoxin alpha (LTα, previously known as TNFβ). Receptor 1 (TNFR1) is involved in pathological processes such as inflammation, apoptosis, and necrosis, while receptor 2 (TNFR2) plays a role in cytotoxicity, host defense response, tissue regeneration, and repair [6].

Taking into account all the roles of TNF-α that have a negative impact, the aims of this study were to assess the transcriptional activity of the studied TNF-α genes and their receptors in patients with various degrees of coronary artery disease and in the control group, as well as to attempt to link it with the size of the left ventricular ejection fraction and the number of diseased coronary arteries.

## 2. Results

### 2.1. Characteristics of the Study Group, Including Laboratory Test Results

The characteristics of the study group of patients, including the laboratory test results, are presented in Table 1.

In the studied group of patients with various degrees of advancement of coronary artery disease, a more unfavorable lipid profile was found (higher concentration of total cholesterol, triglycerides, LDL fraction, and lower concentration of HDL cholesterol) compared to patients with coronary artery disease excluded in coronary angiography (control group). In the remaining patients, abnormal serum glucose levels were also found.

### 2.2. Characteristics of the Study Group with Respect to Transcriptional Activity of Tumor Necrosis Factor Alpha Genes and Their Receptors

The characteristics of the study group of patients with respect to the transcriptional activity of tumor necrosis factor alpha genes and their receptors are presented in Figure 1.

Along with the degree of coronary artery disease, a significant increase in the transcriptional activity of the TNF-α gene was observed in the patients of the study group. The highest transcriptional activity of the receptor 1 and 2 gene (TNFR1, TNFR2) was demonstrated in the control group. A significant decrease in the transcriptional activity of the TNFR1 and TNFR2 genes was observed between the individual groups: the control group and early atherosclerosis, the control group and stable coronary artery disease, as well as between the control group and ACS. Analysis of the transcriptional activity of the TNFR1 and TNFR2 genes did not show statistically significant differences between patients with stable coronary artery disease and the group with acute coronary syndrome.

### 2.3. Characteristics of the Study Group with Stable Coronary Artery Disease and Acute Coronary Syndrome, Taking Into Account the Transcriptional Activity of Tumor Necrosis Factor Alpha Genes and Their Receptors, as Well as the Left Ventricular Ejection Fraction

The characteristics of the study group of patients with stable coronary artery disease and acute coronary syndrome, taking into account the transcriptional activity of tumor necrosis factor alpha genes and their receptors, as well as the left ventricular ejection fraction, are presented in Figure 2.

In the group with stable coronary artery disease and an ejection fraction below 50%, statistically significantly lower transcriptional activity of the TNF-α gene was demonstrated compared to the high activity of the studied gene in patients with acute coronary syndrome and an ejection fraction above 50%. No statistically significant differences were demonstrated in the transcriptional activity of the type I and II receptor genes of tumor necrosis factor alpha between the results with acute coronary syndrome and stable coronary artery disease, regardless of the size of the left ventricular ejection fraction.

### 2.4. Characteristics of the Study Group with Respect to Transcriptional Activity of Tumor Necrosis Factor Alpha Genes and Their Receptors and the Number of Diseased Coronary Arteries

Characteristics of the study group of patients with stable coronary artery disease and acute coronary syndrome with respect to the transcriptional activity of tumor necrosis factor alpha genes and their receptors, and the number of diseased coronary arteries are presented in Figure 3.

Considering the hemodynamically significant stenoses in coronary angiography, the analysis showed the significantly higher transcriptional activity of the TNF-α gene in the group with acute coronary syndrome and multivessel disease compared to patients with stable multivessel coronary disease. No statistically significant differences were found in the transcriptional activity of the tumor necrosis factor alpha receptor type I and II genes in the group of patients with acute coronary syndrome and stable coronary artery disease, regardless of the number of diseased coronary arteries.

Table 2 presents the characteristics of the study group, including the results of the multivariate regression for research variables potentially predicting the value of the transcriptional activity of the TNF-alpha and its receptors gene in patients with early stages of coronary artery disease and stable coronary artery disease. Appendix A present the results of all multivariate analyses.

The performed regression analysis showed that in patients with early stages of CAD, obesity/overweight significantly increased the transcriptional activity of the TNF-α gene. Moreover, in the group of patients with stable coronary artery disease, it was found that the transcriptional activity of the TNF receptor 2 gene was influenced by overweight/obesity and the occurrence of chronic kidney disease.

## 3. Discussion

It should be noted that most of the scientific studies conducted so far concerned the concentration of TNF-α and not, as in this study, the transcriptional activity of the studied genes, which is a process consisting of reading and rewriting the genetic information contained in the gene, the final product of which is a protein [7].

The aforementioned process is complex and subject to regulation at each stage, and the assessment of the transcriptional activity of genes is not easy. The determination of the TNF-α concentration in peripheral blood serum concerns its total concentration, regardless of the source of origin. In this study, the transcriptional activity of the gene was determined in the pool of mononuclear cells (monocytes, lymphocytes), which play a significant role in inflammatory processes.

The inflammatory process is present at all stages of the development and progression of coronary atherosclerosis, and therefore at every stage of the development of coronary artery disease. Metabolic disorders, especially hyperlipidemia and diabetes, as well as hypertension, cause increased oxidative stress leading to vascular endothelial dysfunction, which ultimately leads to the formation of unstable atherosclerotic plaque and its rupture.

Over the past few decades, many innovative molecular studies have been conducted to identify the pathomechanisms of coronary artery disease development, its prevention, and its treatment. Initially, one gene was analyzed. Unfortunately, such studies did not reflect the correct relationships in the body. Thanks to the rapid development of molecular biology techniques, it is now possible to analyze entire signaling pathways. Such a broad view is useful because the signaling cascade is regulated and can be influenced by various stimuli.

Unfortunately, the number of available publications on the assessment of the transcriptional activity of genes, and not their concentration in blood serum, is small. Several studies assessed the transcriptional activity of TNF-α genes in coronary artery disease, but no publications were found that included the assessment of their activity at various stages of the disease, as in the case of the presented results. Due to the similarity in the structure of human and mouse TNF-α (50.4 kDa vs. 50 kDa), scientists began their research on mouse models. Based on independent studies on animals (on mouse models), many researchers have shown a significant effect of blocking the TNF-α gene on delaying the development of atherosclerotic plaque [8,9].

In turn, Park and Heo studied the treatment of atherosclerosis in old mice lacking the low-density lipoprotein receptor. They examined the treatment effect with the following combination of drugs: pravastatin (statin), sarpogrelate (antiplatelet drug), and etanercept (TNF-α inhibitor) and analyzed the development of atherosclerotic plaque in the aorta of mice, and they also measured the expression of the TNF-α gene. The results obtained by the aforementioned subjects were promising (reduction in the area of atherosclerotic lesions and significant reduction in TNF-α expression). The studies conducted suggested that adding a TNF-α inhibitor to conventional therapy may constitute a new treatment strategy for elderly patients with atherosclerosis [10].

The available literature has shown many works on the role of TNF-α polymorphisms and its receptors in coronary artery disease. Sbarsi et al. did not demonstrate any relationship between TNFR1 and TNFR2 receptor polymorphisms and predisposition to CAD. They also suggested that the TNF-308A allele is a predisposing factor and significantly increases the risk of CAD in the presence of comorbidities such as diabetes and hypertension [11]. In turn, Huangfu et al. in their meta-analysis on a group of 16,774 patients (8351 cases and 8423 controls) did not observe any association of the 1031T/C, 857C/T, and 863C/A polymorphisms with an increased risk of CAD [12]. Other researchers agreed with the above-mentioned results. They confirmed the lack of association of the TNF-α gene polymorphisms −308G/A, 857C/T, 863C/A, and 1031 T/C. At the same time, they indicated a significant association of the TNF-α 238G/A polymorphism with the incidence of coronary artery disease in the European and North Asian population [13].

The study conducted by Lu et al. assessed the relationship between serum TNF-α concentration and the severity of coronary artery disease in patients with type II diabetes. They demonstrated a significantly higher level of the concentration of the mentioned cytokine in the group of patients with significant coronary artery stenosis (>70%) compared to the group with normal coronary arteries in coronary angiography. The aforementioned measurements showed a positive correlation with the number of coronary arteries (*p* < 0.01) [14].

Our study demonstrated a significantly higher transcriptional activity of the TNF-α gene in the group of patients with acute coronary syndrome and multivessel disease compared to patients with stable multivessel coronary disease. The observed increase in the transcriptional activity of the studied gene may be caused by the intensification of the inflammatory processes associated with the rupture of the atherosclerotic plaque in acute coronary syndrome.

Enayati et al. did not show a significant statistical difference in the transcriptional activity of the TNF-α gene in patients with coronary artery disease compared to the control group (*p* = 0.980). This study included patients with coronary artery stenosis > 50% (CAD (+)) confirmed by coronary angiography. It should be noted that the group of patients included in this study was small (CAD (+) *n* = 25 people, CAD (−) *n* = 25 people) [15]. Similar conclusions were reached by Hassan-Nejhad et al., who assessed the transcriptional activity of the TNF-α gene in patients with coronary artery disease diagnosed under the age of 50 [16]. Different conclusions were reached by Rafiq et al., who examined the expression of thrombomodulin and TNF-α and NF-KB genes in patients with coronary artery disease in the Pakistani population. The expression of TNF-α and NF-kB genes was significantly lower in patients with CAD. In further studies, they explained the obtained results by the influence of cholesterol-lowering drugs in blood serum on the change in the anti-inflammatory effect of cytokines [17].

The results concerning the transcriptional activity of the genes of type I and II receptors of tumor necrosis factor alpha (TNFR1, TNFR2) were also surprising; they were the highest in the group with coronary atherosclerosis excluded in coronary angiography. This fact could be explained by the lack of specificity of the tested receptors and the increase in their activity in other pathological processes. It is worth mentioning that studies conducted by Rossi et al. on gastric carcinogenesis, which has a similar basis to CAD inflammation, showed that the up-regulation of cell survival genes, e.g., TNFR2 of this signaling pathway in gastric carcinoma, stimulates cell growth, possibly by TNFR2, and negatively controls TNFR1-mediated apoptosis by the down-regulation of pro-apoptotic mediators [18]. Moreover, the mentioned study suggests that the TNF-α signaling pathway can be regulated by the action of miRNAs, mainly miR-19a, which also plays an important role in foam cell formation, responsible for the development of the disease and its distribution, including CAD [19].

In our study, we demonstrated the increasing transcriptional activity of the tested genes with the degree of the advancement of coronary artery disease. Additionally, it was noted that the transcriptional activity of the tested genes was significantly higher in the group with excluded coronary artery disease compared to early and stable coronary artery disease and acute coronary syndrome. There was a significant decrease in activity between the control group and the early atherosclerosis, stable coronary artery disease, and ACS group. Analysis of the transcriptional activity of the TNFR1 and TNFR2 genes did not show statistically significant differences between the group with stable coronary artery disease and patients with acute coronary syndrome.

In this study, the transcriptional activity of the tumor necrosis factor alpha genes and their receptors was also assessed in the studied group of patients with stable coronary artery disease and acute coronary syndrome, taking into account the left ventricular ejection fraction. By definition, the ejection fraction is defined as the ratio of the left ventricular volume at the moment of blood ejection to its end-diastolic volume. A reduced ejection fraction indicates heart failure [20]. In the work on the assessment of TNF-α and its receptors concentration in patients with heart failure with regard to the left ventricular ejection fraction, it was shown that the concentrations of TNF-α and TNFR1 were significantly higher in patients with preserved ejection fraction compared to healthy individuals in the control group. On the other hand, the concentration of TNFR2 was significantly higher in patients with reduced left ventricular ejection fraction compared to the control group [21].

The conducted analysis showed a significantly lower transcriptional activity of the TNF-α gene in patients with stable coronary artery disease with an ejection fraction below 50% compared to a significantly higher transcriptional activity of the studied gene in the group of patients with acute coronary syndrome with an ejection fraction above 50%, which may result from an intensified inflammatory process in the advanced stage of coronary artery disease. Moreover, no statistically significant differences were observed in the transcriptional activity of the receptor genes of the studied cytokine.

The limitations of this study include the small number of study participants and the fact that even the high transcriptional activity of genes is not identical to the concentration of the final product, which is protein, due to the presence of numerous signaling pathways and the co-occurrence of signals from different receptors. In summary, coronary artery disease is a significant epidemiological problem of present times. Moreover, PBMCs are convenient for assessing systemic inflammation, but they may not fully capture the local inflammatory processes occurring in the vascular endothelium, where CAD-related inflammation primarily takes place. On the other hand, they are easily available and constitute good material for conducting research.

The growing number of studies on the transcriptional activity of genes and receptors from the TNF-α family in the future may contribute to a better understanding of their signaling pathways, allowing for the prevention of the development of coronary atherosclerosis and its progression at early stages of development and the introduction of an innovative treatment that prevents the occurrence of life-threatening complications. The ideal targeted drug should have the ability to normalize cytokine signaling pathways while maintaining their basic functions. Therefore, research should be continued to explain these complex mechanisms, and then appropriate therapeutic procedures should be implemented.

## 4. Materials and Methods

This study was initiated after obtaining the consent of the Bioethics Committee of the Medical University of Silesia in Katowice (KNW/0022/KBI/69/18 from 25 September 2018, PCN/0022/KBI/36/21 from 18 May 2021). All patients admitted to the Department of Cardiology Clinic for the diagnostics and treatment of coronary artery disease were analyzed.

The inclusion criteria for this study were being aged over 18 years, providing written, informed consent to participate in the study, and consent to undergo coronary angiography, and the inclusion criterion for the group with stable coronary artery disease and the group with acute coronary syndrome was confirmation of the above diagnoses in imaging studies (coronary angiography, ultrasound), laboratory tests (necrosis markers), ECG (changes in the ST-T segment and T wave), and in the subjective examination (discomfort, chest pain, impaired exercise tolerance).

The exclusion criteria were a lack of patient consent, active inflammatory condition (e.g., infectious endocarditis, pneumonia, exacerbation of chronic obstructive pulmonary disease, etc.), chronic inflammatory diseases (e.g., collagenoses), active neoplastic disease, advanced renal failure, and menstruation in women and difficult contact with the patient (e.g., after stroke).

The above criteria were developed when planning this study and allowed for the separation of the study subgroups. Moreover, the developed exclusion criteria ensured protection of the obtained results against the influence of other diseases that may affect the occurrence of inflammation in the human body.

Taking into account the inclusion and exclusion criteria, a total of 240 people (100%) qualified for this study. All participants gave their informed consent to participate in this study. Among them were 82 (34.17%) women and 158 (65.83%) men. The study participants were aged 29 to 85. The average age was 63.72 ± 10.8 years.

For the proper interpretation of the results of the molecular analyses, the study group (240, 100%) was divided into a control group (C: *n* = 60; 25%), including patients with coronary artery disease excluded in coronary angiography, a group of patients with early coronary artery disease (W: *n* = 60; 25%), a group of patients with stable coronary artery disease (S: *n* = 60; 25%), and a group of patients with acute coronary syndrome (ACS: *n* = 60; 25%).

The transcriptional activity of the TNF-α genes and their receptors was assessed in peripheral blood mononuclear cells. The aforementioned cells were obtained from peripheral blood samples collected simultaneously with blood for other laboratory tests from the basilic vein of the study patients within 48 h of admission to the department. Blood samples were collected in EDTA-coated tubes (Greiner Bio-One, Kremsmünster, Austria) and then centrifuged in a Ficoll gradient to obtain the aforementioned cell population. Ribonucleic acid (RNA) extraction from mononuclear cells was performed using the TRIzol Reagent RNA isolation preparation (Applied Biosystems, Thermo Fisher Scientific, Waltham, Massachusetts, United States of America) based on the modified Chomczyński and Sacchi method. Total RNA extracts were quantitatively assessed by the spectrophotometric measurement of its concentration. The transcriptional activity of the studied genes was assessed using the quantitative real-time polymerase chain reaction method using commercially available analytical kits (Applied Biosystems, Thermo Fisher Scientific, Waltham, Massachusetts, United States of America). The expression of the studied genes was inferred based on the number of mRNA copies per 1 µg of total RNA. Transcriptional activity was analyzed based on the number of mRNA per 1 µg of total RNA according to the available guidelines and with the assumption that there are differences in the concentrations of individual genes in the studied patients, which was an attempt at standardization.

The obtained results were collected in an Excel spreadsheet and exported to Statistica ver. 12.1. Mean values, standard deviation (SD), median (Me), and interquartile range (IQR) were calculated. In order to compare the analyzed parameters in the studied groups, the ANOVA and Kruskal–Wallis tests were used. A significance level of *p* ≤ 0.05 was considered statistically significant.

## 5. Conclusions

The observed increase in the transcriptional activity of the TNF-α gene with a concomitant decrease in the activity of its receptor genes with the advancement of coronary artery disease, compared to the control group, may indicate their significant participation in the development and progression of the disease and constitute a useful marker in non-invasive and early diagnostics. In the patients of the study group, no changes in the transcriptional activity of the TNF-α genes and their receptors were demonstrated depending on the number of diseased coronary arteries and the size of the ejection fraction of the left ventricle.

## Figures and Tables

**Figure 1 ijms-25-13102-f001:**
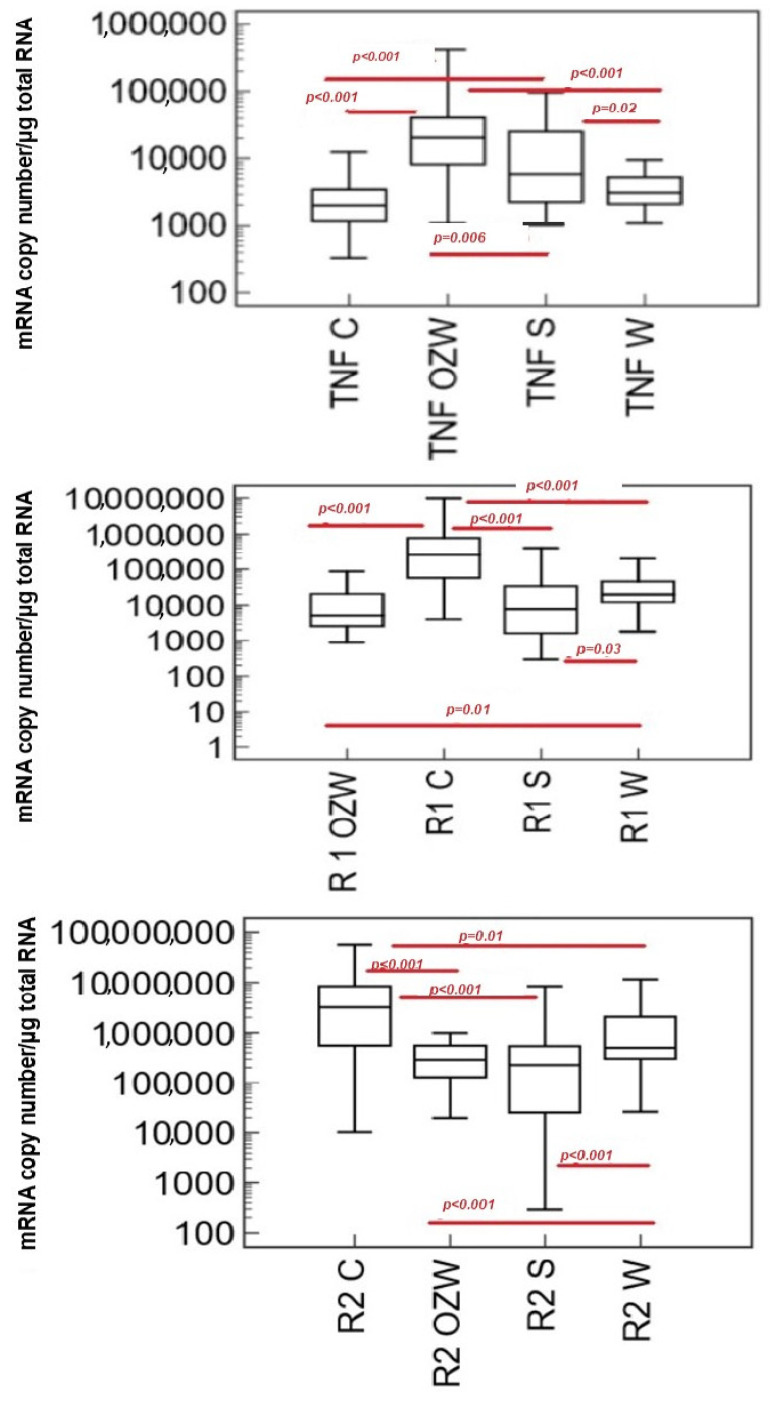
Characteristics of the study group with regard to transcriptional activity of TNF-α, TNFR1, and TNFR2 genes and various degrees of coronary artery disease (Kruskal–Wallis test). Abbreviations: TNF—tumor necrosis factor alpha, R1—tumor necrosis factor alpha type I receptor, R2—tumor necrosis factor alpha type II receptor, C—control group, W—patients with early coronary artery disease, S—group with stable coronary artery disease, OZW—group of patients with acute coronary syndrome, *p*—statistically significant difference.

**Figure 2 ijms-25-13102-f002:**
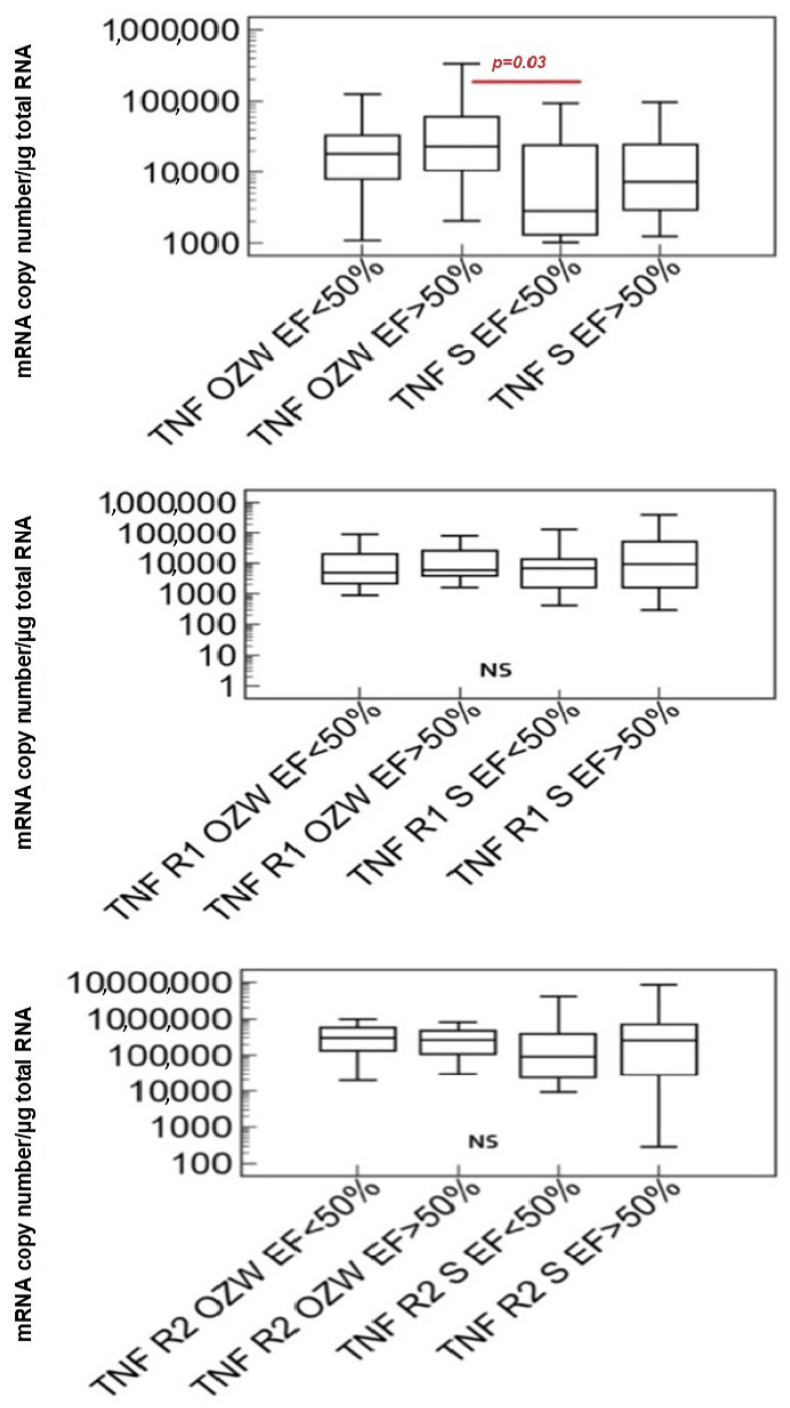
Characteristics of the study group with stable coronary artery disease and acute coronary syndrome, taking into account the transcriptional activity of the TNF-α, TNFR1, and TNFR2 genes and the left ventricular ejection fraction (Kruskal–Wallis test). Abbreviations: TNF—tumor necrosis factor alpha, R1—tumor necrosis factor alpha type I receptor, R2—tumor necrosis factor alpha type II receptor, EF—left ventricular ejection fraction, S—group with stable coronary artery disease, OZW—group of patients with acute coronary syndrome, *p*—statistically significant difference.

**Figure 3 ijms-25-13102-f003:**
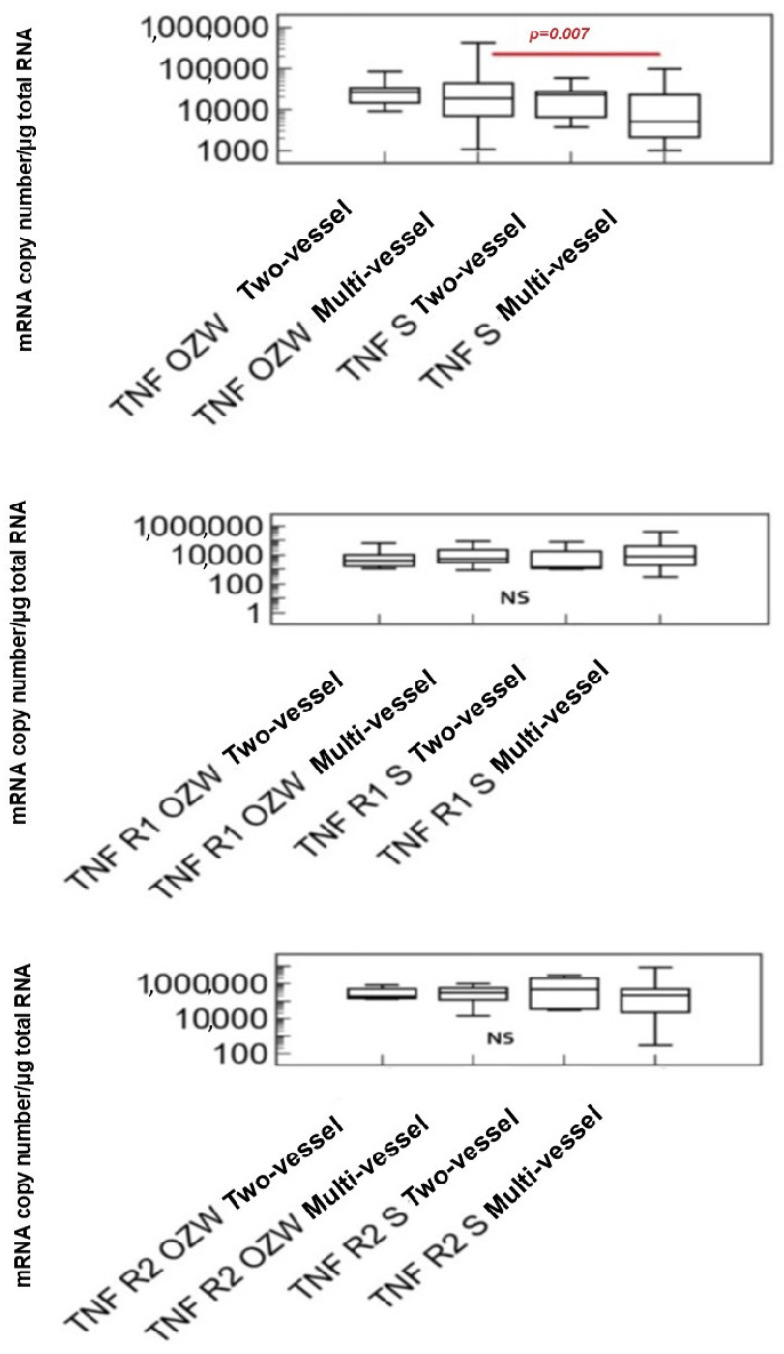
Characteristics of the study group with stable coronary artery disease and acute coronary syndrome, taking into account the transcriptional activity of the TNF-α, TNFR1, and TNFR2 genes and the left ventricular ejection fraction (Kruskal–Wallis test). Abbreviations: TNF—tumor necrosis factor alpha, R1—tumor necrosis factor alpha type I receptor, R2—tumor necrosis factor alpha type II receptor, S—group with stable coronary artery disease, OZW—group of patients with acute coronary syndrome, *p*—statistically significant difference.

**Table 1 ijms-25-13102-t001:** Characteristics of the study group, including laboratory test results.

Variable	Study Group										*p*
	C/W/S/ACS *n* = 240 (100%)		C *n* = 60 (25%)		W *n* = 60 (25%)		S *n* = 60 (25%)		ACS *n* = 60 (25%)		
Tested parameter	X ±SD	Me IQR	X ±SD	Me IQR	X ±SD	Me IQR	X ±SD	Me IQR	X ±SD	Me IQR	
**Total cholesterol [mg/dL]** **(N: 115–190)**	172.97 ± 43.96	167.00 44.00	160.35 ± 27.28	161.00 22.00	181.92 ± 39.02	180.0 0 48.00	164.17 ± 54.72	158.00 37.00	183.13 ± 44.15	185.00 70.50	**<0.001**
**Cholesterol HDL** **[mg/dL**] **(Women N: >45)** **(Men N: >40)**	49.67 ± 21.94	45.00 15.00	60.69 ± 36.86	47.00 12.00	53.58 ± 13.60	50.00 22.00	44.45 ± 7.54	45.00 10.50	40.33 ± 11.34	45.50 12.50	**<0.001**
**Cholesterol LDL** **[mg/dL]**	99.43 ± 35.81	93.00 46.00	81.53 ± 23.42	85.00 33.00	105.75 ± 34.48	102.5 0 44.50	94.92 ± 30.55	90.50 33.50	115.20 ± 43.39	113.00 72.00	**<0.001**
**Triglycerides [mg/dL]** **(N: <150)**	133.87 ± 56.42	125.00 70.00	138.62 ± 39.61	142.50 61.00	119.88 ± 47.54	117.00 60.50	120.58 ± 49.88	99.00 80.00	156.33 ± 74.83	154.00 107.50	**0.002**
**Glucose concentration in blood serum [mg/dL]** **(N: 74–106)**	112.73 ± 35.44	101.00 35.00	93.38 ± 32.36	95.00 10.00	90.05 ± 18.10	94.50 27.50	115.00 ± 34.01	103.00 37.50	134.68 ± 47.55	122.50 40.50	**<0.001**
**Creatinine [mg/dL]** **(N: 0.67–1.17)**	1.36 ± 0.79	0.89 0.27	0.96 ± 0.25	0.93 0.3	0.85 ± 0.21	0.84 0.20	1.08 ±0.74	0.94 0.29	0.93 ± 0.22	0.90 0.27	NS

**Abbreviations:** C—control group, W—patients with early coronary artery disease, S—group with stable coronary artery disease, ACS—group of patients with acute coronary syndrome, X—mean, ±SD—standard deviation, Me—median, IQR—interquartile range, Kruskal–Wallis ANOVA test, N—norm, n—number of patients, *p*—statistical significance, NS—statistically insignificant difference.

**Table 2 ijms-25-13102-t002:** Results of multiple regression for research variables predicting the value of transcriptional activity of the TNF-alpha and its receptors gene in patients with early stages of coronary artery disease and stable coronary artery disease.

Patients With	Early Stages of Coronary Artery Disease					Stable Coronary Artery Disease				
Genes	TNF-α					TNFR2				
Variables	B	SE	Beta	T	*p*	B	SE	Beta	T	*p*
**Age**	−0.004	0.137	−1.093	−0.027	0.978	−0.236	0.145	−51,643	−1.629	0.109
**Family history**	0.267	0.143	1580.018	1.872	0.067	0.252	0.137	1,744,379	1.833	0.073
**Hypertension**	0.021	0.134	239.641	0.153	0.879	0.050	0.137	344,803	0.363	0.718
**Diabetes/Prediabetes**	0.143	0.141	884.509	1.013	0.316	0.053	0.132	203,715	0.398	0.692
**Obesity/Overweight**	−0.338	0.135	−1804.601	−2.502	0.016 *	−0.268	0.130	−557,059	−2.061	0.044 *
**Tobacco smoking history**	−0.050	0.140	−272.356	−0.360	0.721	−0.051	0.136	−199,819	−0.377	0.708
**Chronic kidney disease**	−0.004	0.137	−1.093	−0.027	0.978	0.339	0.137	1,745,418	2.485	0.016 *
**Constans**	R^2^ = 0.11					R^2^ = 0.23				
	Adjusted R^2^ = −0.007					Adjusted R^2^ = 0.12				
	F = 0.93					F = 2.196				
	P = 0.48					P = 0.04				

**Abbreviations:** TNF-tumor necrosis factor alfa, TNFR2—tumor necrosis factor alpha type II receptor, B—unstandardized regression coefficient, SE—standard error, Beta—standardized regression coefficient, T—test statistics, *p*—statistical significance, *—result statistically significant.

## Data Availability

Data are available through the corresponding author upon reasonable request.

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
