# Peer review of "Transcriptional Activity of Tumor Necrosis Factor Alpha Genes and Their Receptors in Patients with Varying Degrees of Coronary Artery Disease"

_ijms, 2024, doi:10.3390/ijms252313102_

Round 1

Reviewer 1 Report

Comments and Suggestions for Authors

1. Consider refining the objectives to emphasize why TNF-α and its receptors are hypothesized to play a critical role in CAD progression. Linking the objectives explicitly to the broader significance of CAD markers in diagnostics could strengthen the introductory section.

2. Provide more context on the inclusion and exclusion criteria to clarify the selection process and representativeness of the study sample. Also, elaborate briefly on why mRNA expression levels per 1µg of total RNA were chosen as a metric, as this would enhance transparency regarding the data interpretation.

3. The authors could enhance this section by exploring potential mechanisms underlying the reduction in receptor gene activity alongside TNF-α upregulation. Additionally, discussing the lack of significant differences in TNF-α receptor gene activity based on ejection fraction and number of affected arteries could strengthen the overall analysis.

4. The study could benefit from a brief section on potential future research, such as investigating other inflammatory markers or expanding the sample size to confirm the findings in diverse populations.

Author Response

Dear Reviewer,

All comments are provided in the attached file.

Your sicnerely,

Reviewer 2 Report

Comments and Suggestions for Authors

This manuscript explores the transcriptional activity of TNF-α and its receptors, TNFR1 and TNFR2, in patients with different stages of CAD. The authors have conducted a substantial study involving 240 patients, which adds significant value to the research. The findings suggest that TNF-α transcriptional activity increases with the progression of CAD, while the activity of its receptors decreases, potentially serving as markers for early, non-invasive diagnosis.

However, the following points need to be addressed:

  1. While PBMCs are convenient for assessing systemic inflammation, they may not fully capture the local inflammatory processes occurring in the vascular endothelium, where CAD-related inflammation primarily takes place. The authors should address this limitation and consider including analyses of vascular tissue samples or endothelial cells to better reflect the true inflammatory state at the lesion sites.
  2. The decrease in TNFR1 and TNFR2 transcriptional activity might be influenced by factors unrelated to CAD, given that these receptors are widely expressed across various cell types and can be affected by other inflammatory or physiological conditions. The authors should clarify how they have accounted for potential confounders to ensure that the observed changes are specifically attributable to CAD progression. Additionally, it may be beneficial to utilize publicly available datasets from resources like the NCBI GEO database (e.g., GSE12288) to corroborate the observed results and strengthen the study's findings.
  3. While you acknowledge in the limitations section that high transcriptional activity of genes does not necessarily equate to protein concentration due to numerous signaling pathways and the co-occurrence of signals from different receptors, this recognition does not suffice as a justification for not assessing protein levels. Evaluating the protein expression of TNF-α and its receptors is essential because mRNA levels alone may not accurately reflect protein abundance or functional activity due to post-transcriptional and translational regulatory mechanisms.
  4. The absence of a significant association between TNF-α and receptor gene expression with left ventricular ejection fraction could be influenced by unaccounted variables such as medication use, comorbidities, or patient history. The authors should control for these potential confounding factors, perhaps through multivariate analysis, to more accurately assess the relationship between gene expression and cardiac function.
  5. To strengthen the causal inference between TNF-α concentration and CAD risk, the authors might consider performing a two-sample Mendelian randomization analysis. This method utilizes genetic variants as instrumental variables to minimize confounding and bias, thereby providing stronger evidence for a causal relationship.

Comments on the Quality of English Language

The authors should also enhance the introduction and methods sections and improve the overall quality of English language usage.

Author Response

(The authors gave the same response as above.)

Round 2

Reviewer 2 Report

Comments and Suggestions for Authors

Dear Authors,

Thank you for your detailed responses to my previous comments. I appreciate your efforts to address the concerns raised. However, I must express that some of your replies do not satisfactorily resolve the issues.

Regarding my second point, while you acknowledge that the decrease in TNFR1 and TNFR2 transcriptional activity might be influenced by factors unrelated to CAD, your explanation falls short of adequately addressing the potential confounders. Merely assuming that all possible additional factors will affect the parameters studied does not provide assurance that the observed changes are specifically due to CAD progression. It is crucial to control for these confounding variables to strengthen the validity of your conclusions. Incorporating statistical methods such as multivariate regression analysis could help account for the influence of medication use, comorbidities, and patient history on gene expression levels.

Furthermore, since your study relies solely on qPCR to assess mRNA expression levels, it is important to corroborate your findings with additional evidence. I strongly recommend performing an analysis using publicly available gene expression datasets, such as those from the NCBI GEO database. By comparing your results with data from other studies or larger cohorts, you can verify whether the observed differences in TNF-α and its receptors are consistent and reproducible. This approach would not only strengthen the credibility of your findings but also address concerns regarding the specificity of the transcriptional changes to CAD.

In addition, integrating data from external sources can provide a broader context for your results and help mitigate the limitations associated with using PBMCs as a proxy for vascular inflammation. Such an analysis could offer valuable insights into whether similar transcriptional patterns are observed in different populations or tissue types, thereby enhancing the generalizability of your study.

I urge you to consider these suggestions seriously. Providing additional database analyses to confirm your findings will significantly improve the robustness and impact of your research. Addressing these concerns will also help ensure that your conclusions are well-supported and that the potential implications for early, non-invasive diagnosis of CAD are justified.

Thank you for your attention to these matters. I look forward to reviewing your revised manuscript.

Comments on the Quality of English Language

NA

Author Response

Dear Reviewer,

All responses were placed in attached file.

Your sincerelly,

Round 3

Reviewer 2 Report

Comments and Suggestions for Authors

Thank you for your response. Aside from a few punctuation errors, there are no further issues.